# Test-Time Adaptive Object Detection with Foundation Model

**Yingjie Gao**[1,2]    **Yanan Zhang**[3]    **Zhi Cai**[1,2]    **Di Huang**[1,2*]

[1]State Key Laboratory of Complex and Critical Software Environment, Beihang University
[2]School of Computer Science and Engineering, Beihang University
[3]School of Computer Science and Information Engineering, Hefei University of Technology
{gaoyingjie, caizhi97, dhuang}@buaa.edu.cn,  yananzhang@hfut.edu.cn

## Abstract

In recent years, test-time adaptive object detection has attracted increasing attention due to its unique advantages in online domain adaptation, which aligns more closely with real-world application scenarios. However, existing approaches heavily rely on source-derived statistical characteristics while making the strong assumption that the source and target domains share an identical category space. In this paper, we propose the first foundation model-powered test-time adaptive object detection method that eliminates the need for source data entirely and overcomes traditional closed-set limitations. Specifically, we design a Multi-modal Prompt-based Mean-Teacher framework for vision-language detector-driven test-time adaptation, which incorporates text and visual prompt tuning to adapt both language and vision representation spaces on the test data in a parameter-efficient manner. Correspondingly, we propose a Test-time Warm-start strategy tailored for the visual prompts to effectively preserve the representation capability of the vision branch. Furthermore, to guarantee high-quality pseudo-labels in every test batch, we maintain an Instance Dynamic Memory (IDM) module that stores high-quality pseudo-labels from previous test samples, and propose two novel strategies-Memory Enhancement and Memory Hallucination-to leverage IDM's high-quality instances for enhancing original predictions and hallucinating images without available pseudo-labels, respectively. Extensive experiments on cross-corruption and cross-dataset benchmarks demonstrate that our method consistently outperforms previous state-of-the-art methods, and can adapt to arbitrary cross-domain and cross-category target data. Code is available at `https://github.com/gaoyingjay/ttaod_foundation`.

## 1 Introduction

As a fundamental task in visual perception, object detection [23, 1, 32] has made significant progress, while its performance drops dramatically when facing domain gaps. Although Unsupervised Domain Adaptation (UDA) technology [3, 34] attempts to mitigate domain differences in an offline manner, it still struggles to meet the real-time domain adaptation requirements in application scenarios such as autonomous driving [33] and robotics [19]. Consequently, Test-Time Adaptation (TTA) [26, 28, 25] has emerged, which operates in real-time by adapting on the fly during inference.

Existing Test-Time Adaptive Object Detection (TTAOD) methods [2, 31, 29], predominantly built upon Faster R-CNN [23], leverage self-training or source-target feature alignment strategies to achieve promising domain adaptation performance. However, as shown in Fig. 1(a), there are two major issues: (1) requiring statistical characteristics (*e.g.*, the mean and variance of feature maps) derived from sampled source domain data, which violates the source-free principle of TTA, and

---

*Corresponding author.

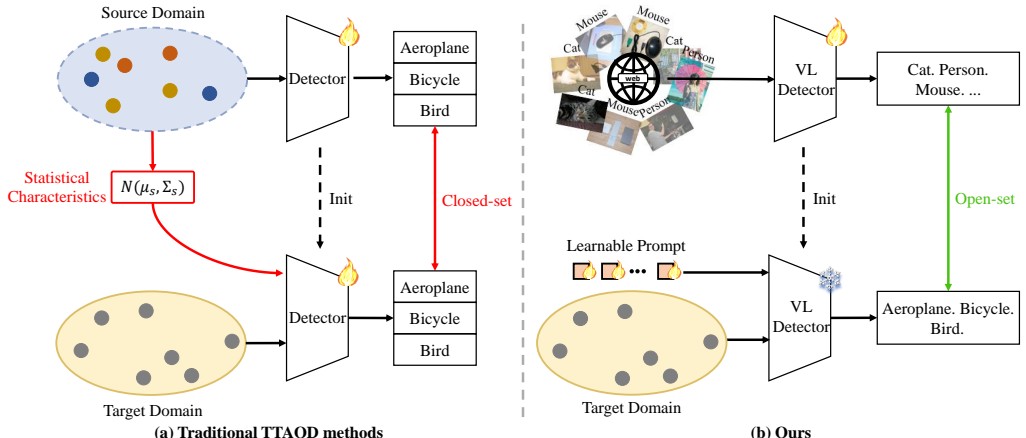

Figure 1: **(a)** Traditional TTAOD methods require source domain statistical characteristics and are limited to closed-set during adaptation. **(b)** Our method requires no source data while possessing open-vocabulary capability.

(2) assuming identical category spaces between source and target domains, which limits TTAOD's applicability in open scenarios.

Recent vision-language foundation models (VLMs) [22, 9, 12, 17], pre-trained on large-scale datasets, have demonstrated remarkable zero-shot generalization and open-vocabulary capabilities, motivating our exploration of introducing vision-language detectors (*e.g.*, GLIP [12] and Grounding DINO [17]) into the TTAOD task to address the aforementioned issues, as shown in Fig. 1(b). However, *how to adapt vision-language detectors during TTA* is non-trivial. On one hand, full-parameter fine-tuning via self-learning on test data not only diminishes the pre-trained detector's generalization capability but also amplifies sensitivity to noisy samples, exacerbating overfitting when target data is scarce. On the other hand, effective adaptation hinges on high-quality pseudo-labels generated from target domain test data, yet consistently obtaining reliable pseudo-labels in every batch remains challenging even with advanced vision-language detectors.

For the first point, a straightforward approach is to perform parameter-efficient fine-tuning on the foundation model using text prompts, but our empirical findings show that tuning only the text prompts is inadequate for effective adaptation. Therefore, we design a Multi-modal Prompt-based Mean-Teacher framework for vision-language detectors to perform self-training during TTA, which incorporates text and visual prompt tuning to jointly adapt both language and vision representation spaces on the test data. Correspondingly, to mitigate potential performance degradation of the teacher model due to suboptimal visual prompt initialization, we introduce a Test-time Warm-start strategy that initializes the visual prompts by average pooling image tokens extracted from the first test sample.

For the second point, we first maintain an Instance Dynamic Memory (IDM) module for each category during test time to help preserve valuable knowledge acquired from prior test samples. Building upon IDM, we then propose two novel strategies: Memory Enhancement and Memory Hallucination. Memory Enhancement leverages high-quality instances stored in IDM to refine the original predictions of the current test image, while Memory Hallucination integrates instances sampled from IDM into test images that have no available pseudo-labels.

The main contributions of this paper are summarized in fourfold:

• To the best of our knowledge, this is the first foundation model-powered test-time adaptive object detector, which eliminates the need for source data entirely and overcomes traditional closed-set limitations.

• We design a Multi-modal Prompt-based Mean-Teacher framework for vision-language detector-driven TTA, which incorporates text-visual prompts and a Test-time Warm-start strategy to achieve effective parameter-efficient fine-tuning while preserving the pre-trained knowledge.

• We introduce an Instance Dynamic Memory module and propose two novel strategies—Memory Enhancement and Memory Hallucination—to effectively leverage high-quality pseudo-labels from previous test samples.

- Extensive experiments on both cross-corruption and cross-dataset benchmarks demonstrate that our proposed method outperforms the state-of-the-art approaches by large margins and can adapt to arbitrary cross-domain and cross-category target data.

## 2 Related Work

### 2.1 Test-Time Adaptive Object Detection

TTAOD extends TTA [7, 26, 28, 8, 25] to the object detection task, aiming to adapt a detector pre-trained on a labeled source domain to different unlabeled target domains in an online manner. Early works [14, 13, 27] employ a self-training paradigm and perform multi-epoch offline adaptation on target domain data. STFAR [2] utilizes self-training to generate pseudo-labeled objects on the fly and incorporates feature distribution alignment as regularization. CTTAOD [31] focuses on continually changing test domains by introducing an adapter-based adaptation approach that activates only when necessary. The latest work, Efficient TTAOD [29], proposes pruning sensitive channels to focus adaptation efforts solely on invariant ones. However, these methods require access to source data for computing statistical characteristics (*e.g.*, mean and variance), which is impractical in real-world scenarios. Moreover, they inherently assume identical category spaces between source and target domains, which significantly limits the applicability of TTAOD. In this paper, we explore leveraging foundation models to enhance TTAOD and overcome the above limitations.

### 2.2 Vision-Language Object Detection

Vision-language foundation models are trained on large-scale image-text pairs collected from the web, which establish connections between visual and textual representations and achieve impressive zero-shot performance on various downstream tasks. Early vision-language object detection works [6, 35] distill knowledge from pre-trained vision-language classification models (*e.g.*, CLIP [22]) to a student detector (*e.g.*, Faster R-CNN [23]), enabling the detection of novel categories beyond the training set. GLIP [12] introduces a grounded language-image pre-training framework that generates grounding boxes in a self-training paradigm, achieving strong zero-shot performance across various object detection datasets. Grounding DINO [17] incorporates grounded pre-training into the Transformer-based detector DINO [32] using a tight cross-modality fusion solution and demonstrates superior generalization ability. In this paper, we investigate how to adapt vision-language detectors during test time and propose an effective and efficient test-time adaptive object detection approach based on Grounding DINO.

## 3 Methodology

### 3.1 Preliminary

Standard TTAOD approaches typically assume access to a detector pre-trained on source domain data $D_S = \{(x_i, y_i)\}_{i=1}^{N_s}$, where $x_i \sim P_S(x)$ and $y_i = (bbox_i, c_i)$ consists of a set of bounding boxes $bbox_i$ and their corresponding class labels $c_i \in C$. The goal of TTAOD is to adapt the detector to different unlabeled target domains $D_T = \{x_j\}_{j=1}^{N_t}$ during testing, where $x_j \sim P_T(x)$ and $P_S(x) \neq P_T(x)$. Crucially, the source domain is unavailable during adaptation, and the target domains share the same label space $C$ with the source domain (*i.e.*, a closed-set scenario). When vision-language detectors are introduced into TTAOD, their large-scale pre-training enables superior generalization on target domains. Moreover, vision-language foundation models break the closed-set constraint, allowing adaptation to arbitrary cross-domain and cross-category target data.

### 3.2 Overall Architecture

An overview of our method is illustrated in Fig. 2. We build our approach on the pre-trained vision-language model Grounding DINO. In Sec 3.3, we introduce a Multi-modal Prompt-based Mean-Teacher framework that enables parameter-efficient self-training on test data. To mitigate potential performance degradation when incorporating visual prompts, we propose a Test-time Warm-start strategy specifically for prompt initialization. In Sec 3.4, we present an Instance Dynamic Memory module to store high-quality pseudo-labels extracted from the test stream, coupled with

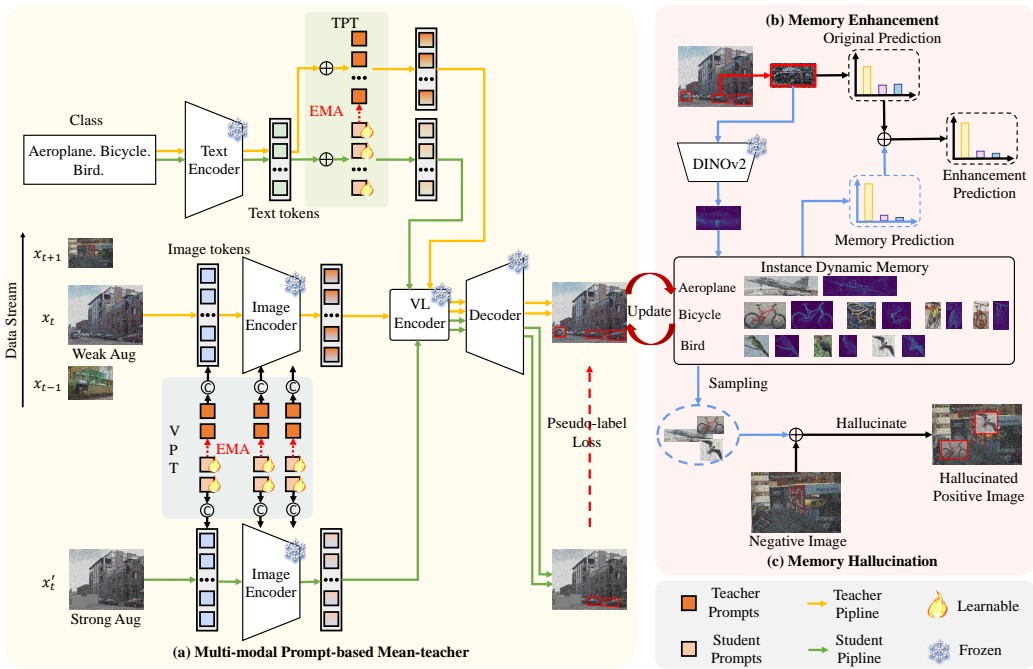

Figure 2: Overview of our method. It comprises two components: (1) the Multi-modal Prompt-based Mean-Teacher framework shown in (a), incorporating text prompt tuning (green-highlighted) and visual prompt tuning (blue-highlighted) with a Test-time Warm-start strategy; and (2) an Instance Dynamic Memory module that stores high-quality pseudo-labels from previous test samples, integrating with Memory Enhancement (b) and Memory Hallucination (c).

two novel strategies—Memory Enhancement and Memory Hallucination—to refine the original predictions for test images and hallucinate positive samples for images lacking reliable pseudo-labels, respectively.

## 3.3 Multi-modal Prompt-based Mean-Teacher

Although vision-language pre-trained detectors like GLIP[12] and Grounding DINO[17] demonstrate impressive zero-shot generalization, their performance often degrades under real-world distribution shifts. Fine-tuning vision-language detectors to target domains during test time is therefore essential. To achieve effective and efficient adaptation while preserving valuable pre-trained knowledge, we propose a Multi-modal Prompt-based Mean-Teacher framework, which primarily comprises three core components: Text Prompt Tuning, Visual Prompt Tuning, and Test-time Warm-start.

**Text Prompt Tuning.** Text prompt tuning is one of the most prominent approaches in parameter-efficient fine-tuning for VLMs[37, 36, 12]. Accordingly, given a test label set $C$, Grounding DINO concatenates the class names in $C$ using dot symbols to form the text input $t$ (*e.g.*, "aeroplane. bicycle. bird. ..."). The text encoder $f_T$ then maps the input description $t$ to a sequence of at most 256 tokens, $E_T$. We further introduce a learnable vector $P_T$ in the language branch, whose dimension matches that of $E_T$. The modulated text tokens are computed as:

$$\tilde{E}_T = E_T + P_T \tag{1}$$

This enriched representation $\tilde{E}_T$ is subsequently fed into the vision-language feature enhancer $f_{VL}$.

**Visual Prompt Tuning.** Observing that fine-tuning the text prompts alone fails to effectively adapt to test data. As shown in Fig. 3(b), we further introduce $m$ learnable tokens $P_{I,i} = \{P_{I,i}^k \in \mathbb{R}^{d_i}\}_{k=1}^{m}$ at each image encoder layer $L_i$ in Grounding DINO, alongside the input image tokens $E_{I,i}$, where $d_i$ denotes the dimension of $L_i$. Similar to [10], we concatenate the visual prompts $P_I$ with the image tokens as the input of each Transformer layer:

$$[\_, \tilde{E}_{I,1}] = L_1([P_{I,0}, E_{I,0}]) \tag{2}$$

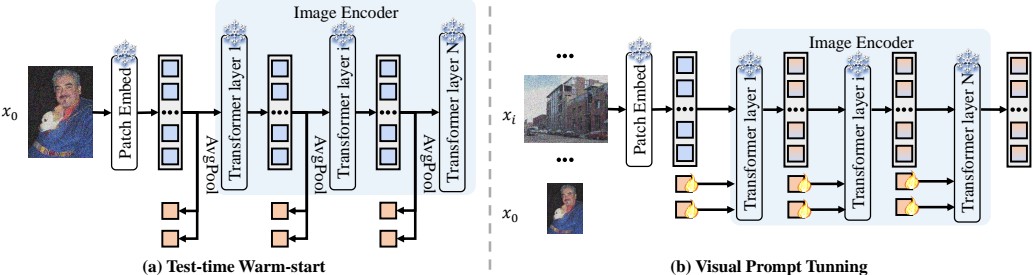

**(a) Test-time Warm-start**        **(b) Visual Prompt Tunning**

Figure 3: Different behaviors of visual prompts. **(a)** Init visual prompts by average-pooling image tokens from the first test sample before TTA. **(b)** Insert visual prompts with image tokens for every test sample during TTA.

$$[\_, \tilde{E}_{I,i}] = L_i([P_{I,i-1}, \tilde{E}_{I,i-1}]) \quad i = 2, 3, \dots, N \tag{3}$$

The visual prompt augmented image tokens $\tilde{E}_{I,N}$ then serve as the visual input to $f_{VL}$.

**Test-time Warm-start.** Here, a challenge remains regarding the initialization of multi-modal prompts. For the text prompts, we simply set $P_T = 0$ so that the output tokens from the text branch remain unchanged. However, for the visual prompts, both zero initialization and random initialization inevitably degrade the representation capability of the vision branch. In extreme cases, the detector fails to detect any objects, resulting in catastrophic failure during test-time adaptation. As illustrated in Fig. 3(a), we propose a Test-time Warm-start strategy for the visual prompts. At the beginning of each test-time adaptation, we initialize the visual prompts using average pooling over the input image tokens $E_{I,i}$ from the first image $X_0$ at the $i$-th Transformer layer:

$$P_{I,i} = AvgPool(E_{I,i}) \tag{4}$$

Based on the above components, we construct the Multi-modal Prompt-based Mean-Teacher framework, which comprises a set of multi-modal teacher prompts $P^*_{\{T,I\}}$ and a set of multi-modal student prompts $P_{\{T,I\}}$. The teacher model, equipped with $P^*_{\{T,I\}}$, generates high-quality pseudo-labels on weakly augmented target domain test data to supervise the optimization of the student model, which uses $P_{\{T,I\}}$ and receives strongly augmented test data. To prevent being misled by noisy pseudo-labels, we set a classification score threshold $th_{pl}$ to filter them. The total optimization objective is:

$$L_{total} = L_{cls} + L_{loc} \tag{5}$$

where $L_{cls}$ and $L_{loc}$ denote the contrastive classification loss and localization loss in Grounding DINO, respectively. Pseudo-labels generated by the teacher model are used as ground-truth during test time. We only fine-tune $P_{\{T,I\}}$, while the majority of the pre-trained detector remains frozen during adaptation. And $P^*_{\{T,I\}}$ are updated after each iteration via exponential moving average of $P_{\{T,I\}}$ as follows:

$$P^*_{\{T,I\}} = \gamma P^*_{\{T,I\}} + (1-\gamma)P_{\{T,I\}} \tag{6}$$

where $\gamma \in [0,1]$ is the momentum coefficient.

### 3.4 Instance Dynamic Memory Enhancement and Hallucination

The mean-teacher framework primarily relies on high-quality pseudo-labels derived from target domain test data to facilitate test-time adaptation. However, obtaining reliable pseudo-labels in every test batch remains challenging, even with the assistance of the vision-language detector. To address this issue, we propose maintaining an Instance Dynamic Memory (IDM) module for each category during TTA to preserve valuable knowledge acquired from prior test samples.

Specifically, the IDM maintains a dynamic queue $Q_c$ for each category $c$, which is initialized as empty. For the current test data $x_i$, one of its pseudo-labels is typically represented as $(bbox, s, c)$, corresponding to the predicted bounding box, classification score, and category, respectively. For a high-quality pseudo-label with classification score $s$ exceeding the given threshold $th_{pl}$, we construct a triplet $(img, feat, s)$ as follows:

$$img = Crop(x_i, bbox) \tag{7}$$

$$feat = DINOv2(img) \tag{8}$$

where $Crop(\cdot, \cdot)$ is used to extract an instance crop from image $x_i$, and $DINOv2(\cdot)$ generates the DINOv2[21] feature for the cropped instance. We then store the triplet in the dynamic queue $Q_c$ corresponding to its predicted category $c$. When $Q_c$ has not reached the maximum capacity $|Q_c|_{max}$, we directly insert this new triplet. However, if $Q_c$ is already full, we compare the classification score $s$ of the current pseudo-label instance with the lowest score in $Q_c$. If $s$ is higher, we replace the lowest-scoring triplet with the new one; otherwise,the new pseudo-label instance is excluded, and the queue keeps unchanged.

IDM maintains progressively improving high-quality pseudo-labels through its iterative refinement mechanism. Based on IDM, we introduce two novel strategies-Memory Enhancement and Memory Hallucination-to more effectively leverage high-quality pseudo-labels from the test stream.

**Memory Enhancement.** Inspired by TDA[11], we leverage the IDM to refine the original predictions of the current test image through the Memory Enhancement strategy, as illustrated in Fig. 2(b). We first calculate the DINOv2 feature prototypes for each class $c$ by averaging all samples in $Q_c$, denoted as $v_c$. Then, for an original prediction $(bbox_j, s_j, c_j)$ of the current test image $x_i$, we compute its memory-based classification score $s_j'$ as:

$$s_j' = \mathcal{A}(DINOv2(Crop(x_i, bbox_j))v_c^T) \tag{9}$$

where $\mathcal{A}(x) = \alpha exp(-\beta(1-x))$ is the affinity function , $\alpha$ is a weighting factor and $\beta$ is a sharpness ratio. The enhanced prediction is defined as $((bbox_j, s_j'', c_j''))$, where the final classification score is $s_j'' = s_j + s_j'$, and its predicted category is $c_j'' = argmax s_j''$.

We compute similarity using the prototype $v_c$ for each category $c$ instead of all samples in $Q_c$, since the number of instances per category varies greatly in object detection. Directly using all samples from each category would cause categories with more instances to obtain higher classification scores. Given that the detector produces numerous predictions per image (*e.g.*, 300 in Grounding DINO), applying Memory Enhancement to every prediction would not only significantly reduce inference speed, but also degrade detection performance due to the enhancement of low-confidence predictions. Thus, we apply Memory Enhancement only to high-confidence predictions whose classification scores exceed $th_{me}$.

**Memory Hallucination.** The threshold filtering in the mean-teacher framework may lead to no available pseudo-labels for adaptation on certain test data, a limitation overlooked by previous self-training based works[27, 2]. Since these negative test samples still contain valuable target domain information, we propose a Memory Hallucination strategy, as shown in Fig. 2(c). This strategy randomly samples high-quality instances from IDM and hallucinates positive samples by integrating them into the negative test data.

Specifically, for a negative image $x_i$ and a high-quality instance image $img_j$ sampled from IDM, we overlay $img_j$ onto $x_i$ at a random position to generate a hallucinated positive image $\tilde{x}_i$ using a mixing coefficient $\lambda$, where $\lambda \in [0, 1]$ is sampled from a Beta distribution. For each negative image, we commix at most three high-quality instance images. To prevent overlap among instances, we set an IoU threshold $th_{IoU}$: if the IoU of the current instance image and any previously placed one exceeds $th_{IoU}$, we randomly reselect positions and retry up to 10 times. Additionally, to prevent the detector from overfitting to high-quality instance images in the scale space, we apply random scaling to $img_j$ before mixing.

## 4 Experiments

### 4.1 Datasets

We evaluate the effectiveness of our method across a variety of TTAOD scenarios, covering two benchmarks: the cross-corruption benchmark and the cross-dataset benchmark. The cross-corruption benchmark is widely adopted in previous TTAOD works[2, 31, 11] to assess model robustness, specifically including two datasets: Pascal-C and COCO-C. **Pascal-C** is constructed from the test set of Pascal VOC[5] by applying an image corruption package [20], which consists of 15 types of corruptions. Each corrupted test set contains 4956 images spanning 20 classes. **COCO-C** is generated from COCO [16], which contains 80 object categories. Following the same procedure as

Table 1: Test-time adaptive object detection results (AP50) on **Pascal-C**.

| Detectors | Methods | Noise | | | Blur | | | | Weather | | | | Digital | | | | Avg |
|---|---|---|---|---|---|---|---|---|---|---|---|---|---|---|---|---|---|
| | | Gauss | Shot | Impul | Defoc | Glass | Motn | Zoom | Snow | Frost | Fog | Brit | Contr | Elast | Pixel | Jpeg | |
| Faster RCNN (ResNet-50) | Direct Test | 11.9 | 16.0 | 13.6 | 16.7 | 13.0 | 18.4 | 25.7 | 38.2 | 41.7 | 64.2 | 69.5 | 23.8 | 42.7 | 26.0 | 35.8 | 30.5 |
| | BN [7] | 4.7 | 6.8 | 5.1 | 7.4 | 4.5 | 9.8 | 13.4 | 19.1 | 22.1 | 35.3 | 39.5 | 20.6 | 17.1 | 9.1 | 10.5 | 15.0 |
| | TENT [28] | 3.1 | 4.0 | 3.3 | 2.6 | 2.5 | 5.3 | 4.8 | 12.8 | 13.7 | 19.0 | 19.6 | 9.9 | 11.0 | 8.8 | 4.5 | 8.3 |
| | T3A [8] | 6.1 | 8.4 | 6.5 | 11.0 | 6.4 | 10.1 | 13.8 | 16.8 | 20.6 | 32.7 | 36.9 | 12.5 | 19.7 | 13.2 | 14.8 | 15.3 |
| | SHOT [15] | 12.0 | 19.9 | 16.4 | 18.9 | 11.6 | 19.7 | 27.6 | 42.5 | 45.8 | 67.5 | 72.0 | 31.7 | 46.6 | 33.1 | 41.8 | 33.8 |
| | Mean-Teacher[30] | 24.8 | 29.0 | 26.5 | 21.7 | 18.9 | 24.8 | 27.7 | 46.1 | 50.5 | 67.8 | 71.4 | 37.3 | 52.7 | 39.7 | 51.1 | 39.3 |
| | STFAR [2] | 29.8 | 38.0 | 34.9 | 30.8 | 31.5 | 32.8 | 29.3 | 51.4 | 53.1 | 68.3 | 71.4 | 47.9 | **58.4** | **48.9** | 50.8 | 45.2 |
| Grounding DINO (Swin-T) | Direct Test | 31.8 | 38.6 | 35.7 | 43.3 | 20.1 | 36.4 | 33.1 | 59.2 | 64.8 | 75.9 | 75.7 | 54.8 | 42.1 | 10.3 | 49.7 | 44.8 |
| | Mean-Teacher[30] | 42.7 | 48.1 | 46.6 | **48.4** | 30.0 | 45.6 | 37.1 | 64.6 | 67.1 | 75.5 | 75.8 | 64.4 | 49.3 | 19.4 | 58.3 | 51.5 |
| | Ours | **46.9** | **52.0** | **51.9** | 47.9 | **38.6** | **48.6** | **39.4** | **66.7** | **68.6** | **77.7** | **77.8** | **66.5** | 54.2 | 42.8 | **63.7** | **56.2** |

Table 2: Test-time adaptive object detection results (mAP) on **COCO-C**. * indicates methods using SoftTeacher weights pre-trained on the COCO training set with extensive data augmentation.

| Detectors | Methods | Noise | | | Blur | | | | Weather | | | | Digital | | | | Avg |
|---|---|---|---|---|---|---|---|---|---|---|---|---|---|---|---|---|---|
| | | Gauss | Shot | Impul | Defoc | Glass | Motn | Zoom | Snow | Frost | Fog | Brit | Contr | Elast | Pixel | Jpeg | |
| Faster RCNN (ResNet-50) | Direct Test | 8.2 | 10.0 | 9.1 | 12.9 | 4.7 | 9.1 | 4.9 | 19.8 | 24.0 | 38.9 | 38.4 | 22.9 | 16.5 | 6.2 | 13.2 | 15.9 |
| | BN [7] | 1.4 | 1.8 | 1.5 | 1.7 | 0.8 | 1.8 | 2.0 | 5.8 | 8.3 | 13.6 | 15.2 | 3.4 | 7.3 | 2.2 | 3.1 | 4.7 |
| | TENT [28] | 1.5 | 1.7 | 1.6 | 0.5 | 0.5 | 1.6 | 0.8 | 5.4 | 6.4 | 9.7 | 8.5 | 5.6 | 5.0 | 2.4 | 2.2 | 3.6 |
| | T3A [8] | 4.6 | 5.8 | 5.2 | 8.3 | 3.1 | 5.8 | 3.5 | 13.8 | 17.2 | 28.9 | 28.8 | 15.9 | 11.3 | 4.1 | 9.0 | 11.0 |
| | SHOT [15] | 11.0 | 13.0 | 12.1 | 14.7 | 7.2 | 11.0 | 6.4 | 22.0 | 26.7 | 41.5 | 40.9 | 26.6 | 19.7 | 9.7 | 16.4 | 18.6 |
| | Mean-Teacher* [30] | 12.3 | 12.6 | 13.6 | 14.4 | 8.7 | 11.9 | 5.9 | 25.1 | 27.0 | 38.5 | 37.8 | 28.7 | 21.2 | 1.5 | 19.3 | 18.6 |
| | STFAR* [2] | 14.8 | 16.7 | 16.5 | 15.1 | 13.4 | 14.1 | 7.5 | 26.5 | 27.2 | 38.5 | 38.4 | 29.2 | 26.3 | 18.5 | 22.4 | 21.7 |
| | CTTOD* [31] | 14.9 | 17.0 | 15.9 | 14.1 | 12.4 | 13.7 | 7.7 | 25.5 | 27.6 | 39.4 | 38.8 | 29.3 | 27.7 | **26.3** | 24.8 | 22.3 |
| | CTTOD-Skip* [31] | 14.3 | 16.2 | 15.3 | 14.2 | 11.9 | 13.2 | 7.3 | 24.0 | 26.9 | 39.0 | 38.9 | 28.3 | 26.2 | 25.4 | 23.7 | 21.7 |
| Faster RCNN (ResNet-101) | Direct Test | 11.7 | 13.8 | 12.2 | 15.1 | 7.1 | 10.9 | 5.5 | 23.3 | 26.9 | 42.5 | 41.8 | 26.8 | 18.9 | 8.7 | 16.0 | 18.7 |
| | Mean-Teacher* [30] | 16.7 | 20.4 | 20.1 | 17.3 | 15.8 | 15.9 | 7.5 | 29.5 | 30.7 | 42.6 | 41.4 | 33.1 | 24.8 | 13.3 | 22.0 | 23.4 |
| | STFAR* [2] | 20.1 | 19.3 | 20.7 | 17.0 | **16.6** | **17.1** | 8.6 | 30.6 | 31.2 | 42.1 | 41.7 | **33.8** | 29.6 | 26.1 | 25.3 | 25.3 |
| Faster RCNN (Swin-T) | Direct Test | 9.7 | 11.4 | 10.0 | 13.4 | 7.5 | 12.1 | 5.2 | 20.7 | 24.8 | 36.1 | 36.0 | 12.9 | 19.1 | 4.9 | 15.8 | 16.0 |
| | CTTOD [31] | 13.5 | 15.8 | 15.1 | 14.3 | 14.2 | 14.9 | **8.8** | 25.1 | 27.2 | 37.6 | 37.0 | 27.5 | 28.6 | 25.2 | 23.7 | 21.9 |
| | CTTOD-Skip [31] | 13.6 | 15.6 | 14.8 | 14.3 | 13.6 | 14.3 | 7.8 | 24.0 | 26.7 | 37.5 | 36.8 | 27.0 | 27.3 | 23.7 | 22.6 | 21.3 |
| Grounding DINO (Swin-T) | Direct Test | 13.7 | 16.0 | 15.0 | 16.8 | 7.5 | 13.6 | 6.7 | 27.5 | 32.5 | 44.2 | 44.1 | 21.9 | 22.5 | 5.3 | 21.1 | 20.6 |
| | Mean-Teacher[30] | 18.6 | 20.8 | 20.4 | **18.4** | 11.0 | 16.6 | 7.6 | 30.9 | 34.6 | **45.3** | 44.8 | 28.8 | 25.8 | 12.3 | 25.7 | 24.1 |
| | Ours | **20.2** | **22.0** | **21.4** | 17.8 | 14.5 | 16.9 | 7.9 | **31.1** | **34.7** | 45.1 | **44.9** | 30.6 | **29.9** | 23.6 | **29.2** | **26.0** |

Pascal-C, we construct COCO-C using the COCO val2017 set, which includes 5k images, to serve as the target domains.

We adopt the **ODinW-13** datasets as a novel cross-dataset benchmark to evaluate the detector's performance across 13 diverse object detection datasets, each representing a distinct domain with different categories. These datasets are labeled as Ae (Aerial Maritime Drone), Aq (Aquarium), Co (Cottontail Rabbits), Eg (Egohands), Mu (Mushrooms), Pa (Packages), Pv (Pascal VOC), Pi (Pistols), Po (Pothole), Ra (Raccoon), Sh (Shellfish), Th (Thermal Dogs and People), Ve (Vehicles). We perform test-time adaptation on the test sets of 13 sub-datasets, providing a comprehensive evaluation of the model's adaptability across varying class spaces.

## 4.2 Implementation Details

In this paper, our method is built upon Grounding DINO with Swin-Tiny[18] as the visual backbone. We use Grounding DINO pre-trained on Objects365[24], GoldG[12], and Cap4M[12], without any fine-tuning on source domain data before adaptation. Additionally, we employ DINOv2 with ViT-L[4] as the feature extractor in the IDM module. For the cross-corruption benchmark, we set the learning rate of the AdamW optimizer to 0.02 for text prompts and 0.2 for visual prompts, while freezing all other parameters pre-trained on large-scale data. The batch size is set to 4. For other hyperparameters, we set $th_{pl}$ to 0.3, $th_{me}$ to 0.3, and $th_{IoU}$ to 0.2. The momentum coefficient $\gamma$ in Eq. 6 is set to 0.999. The number $m$ of visual prompts is set to 10, and the maximum capacity $|Q_c|_{max}$ of IDM is set to 20. We set $\alpha = 5.0$ and $\beta = 5.0$ for Pascal-C, while $\alpha = 1.0$ and $\beta = 5.0$ for COCO-C. All experiments are conducted on a single RTX 3090 GPU. Hyperparameters for the cross-dataset benchmark are detailed in the *appendix*.

## 4.3 Comparisons with State-of-the-art

**Results on the Cross-corruption Benchmark.** We first compare our method with existing TTAOD approaches on Pascal-C. As shown in Table 1, adapting BN[7], TENT[28] and T3A[8] to TTAOD leads to significant performance degradation compared to directly testing using a Faster R-CNN trained on the source domain. In contrast, self-training based methods such as SHOT[15], Mean-Teacher and STFAR yield performance gains. When tested directly, Grounding DINO achieves an

average AP50 of $44.8\%$, comparable to the previous state-of-the-art STFAR. This demonstrates the strong generalization capability of the pre-trained vision-language detector. Furthermore, simply applying self-training to Grounding DINO results in a $6.7\%$ improvement in average AP50. In comparison, our proposed method achieves the highest average AP50 of $56.2\%$ across 15 corruption types, outperforming the previous state-of-the-art STFAR by a remarkable margin of $11.0\%$. Considering the differences in the adopted detectors, achieving a completely fair comparison is challenging. Nevertheless, by conducting thorough comparisons with the strong Mean-Teacher baseline built upon Grounding DINO ($51.5\%$ *vs.* $56.2\%$), we clearly demonstrate the consistent performance improvements achieved by our method.

We then report the mAP performance on COCO-C in Table 2, presenting comprehensive comparisons with conventional TTAOD approaches across different backbones. Since CTTOAD and CTTAOD-Skip are specifically designed for continuous test-time adaptation (CTTA), we evaluate their performance under the discrete adaptation setting on COCO-C. Methods using ResNet-50 and ResNet-101 are initialized with SoftTeacher[30] weights pre-trained on the COCO training set. Since SoftTeacher employs extensive data augmentation during training (*e.g.*, brightness jitter, contrast jitter), it demonstrates superior adaptation capability on corrupted target domains. Our method achieves the highest average mAP of $26.0\%$ without utilizing any source domain data, while attaining state-of-the-art performance on 8 out of 15 corruption types.

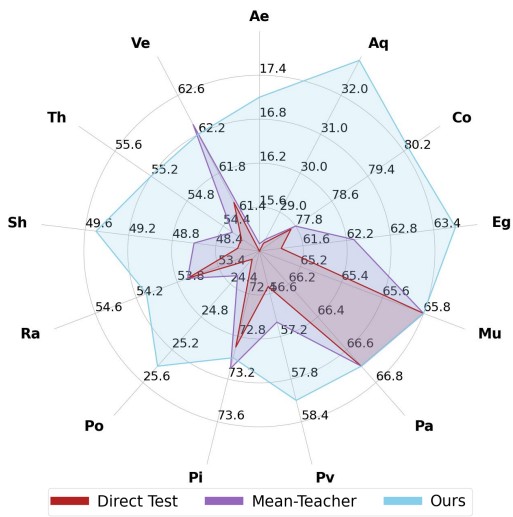

Figure 4: Results on the cross-dataset benchmark comprising 13 diverse object detection datasets.

**Results on the Cross-dataset Benchmark.** Fig. 4 shows our method's performance on 13 downstream datasets with different categories. Our method reaches an average mAP of $54.2\%$, representing a $1.4\%$ improvement over Direct Test's $52.8\%$, while Mean-Teacher with Grounding DINO yields only a $0.3\%$ gain. Our method demonstrates consistent improvements on almost all ODinW-13 sub-datasets except Mu and Pa. This phenomenon is primarily attributed to the extremely few test sanples available (5 for Mu, 4 for Pa), which prevents the vision-language detector from adequately adapting to the target domains. Experimental results verify the effectiveness of our method in adapting to diverse class datasets during test time.

## 4.4 Ablation Studies

**Effectiveness of Each Component.** We investigate the contribution of each component on Pascal-C across all 15 corruption types, with results presented in Table 3. Using only Text Prompt Tuning (TPT) yields merely a $0.6\%$ improvement in average AP50 compared to direct testing. When applying Visual Prompt Tuning (VPT) alone for TTA, the performance even drops by $3.4\%$. We attribute this decline to suboptimal visual prompt initialization, which impairs the teacher model's performance and consequently undermines the adaptation capability of the vision-language detector during test time. Comparing (2) and (3), our proposed Test-time Warm-start (TTWS) strategy

Table 3: Ablation study of each component on Pascal-C. The average AP50 across 15 corruption types is shown.

| Methods | MPMT | | | IDM | | Avg |
|---|---|---|---|---|---|---|
| | TPT | VPT | TTWS | ME | MH | |
| Direct Test | | | | | | 44.8 |
| (1) | ✓ | | | | | 45.4 |
| (2) | | ✓ | | | | 41.4 |
| (3) | | ✓ | ✓ | | | 53.4 |
| (4) | ✓ | ✓ | ✓ | | | 53.9 |
| (5) | | | | ✓ | | 46.1 |
| (6) | ✓ | ✓ | ✓ | | ✓ | 54.9 |
| Ours | ✓ | ✓ | ✓ | ✓ | ✓ | **56.2** |

effectively mitigates this issue. Furthermore, the text and visual prompts exhibit complementary effects. As a training-free strategy, Memory Enhancement (ME) directly enhances the zero-shot performance of Grounding DINO on test data. Comparing (4) and (6), Memory Hallucination

(MH) improves adaptation by hallucinating positive samples on challenging images, enabling the Multi-modal Prompt-based Mean-Teacher (MPMT) to achieve an additional $1.0\%$ performance gain. By integrating all components, our method achieves state-of-the-art performance.

Table 4: Comparison on the Number of Visual Prompts.

| # Prompts | 2 | 4 | 6 | 8 | 10 | 15 | 20 | 30 | 50 |
|---|---|---|---|---|---|---|---|---|---|
| AP50 | 42.6 | 44.0 | 45.0 | 45.3 | 45.4 | 45.2 | 45.3 | 45.1 | 44.2 |

**About the Number of Visual Prompts.** We evaluate the impact of the number $m$ of visual prompts on the Gaussian noise corruption of Pascal-C. As shown in Table 4, the visual-language detector achieves effective test-time adaptation with only a few visual prompts. While increasing the number of visual prompts enhances the detector's adaptability, it also raises the risk of overfitting. For example, performance begins to decrease when $m$ exceeds 30. Empirically, setting $m$ to 10 provides an optimal balance.

**About the Maximum Capacity of IDM.** We analyze the influence of the maximum capacity $|Q_c|_{max}$ of IDM under Gaussian noise corruption on Pascal-C, which simultaneously affects the effectiveness of both the Memory Enhancement and Memory Hallucination strategies. As shown in Fig. 5, when $|Q_c|_{max}$ is set too low, Memory Enhancement fails to acquire sufficiently representative category prototypes $v_c$, thereby limiting its ability to refine the original predictions. When set too high, noisy pseudo-labels may be included, compromising the quality of $v_c$ and weakening the effectiveness of Memory Enhancement. Furthermore, experiments reveal that the introduction of category prototypes $v_c$ confers robustness to Memory En-

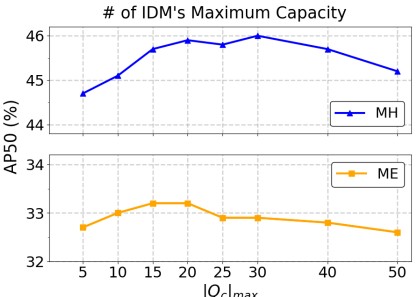

Figure 5: Comparison on the Maximum Capacity of IDM.

hancement. For Memory Hallucination, a too-small $|Q_c|_{max}$ leads to repeated use of few-shot images from IDM for negative image hallucination, causing the detector to overfit and even degrading its performance. Conversely, a large $|Q_c|_{max}$ risks using noisy pseudo-labels during the hallucination process, misleading test-time adaptation. Considering both the performance of the two strategies and the storage cost, we set $|Q_c|_{max}$ to 20.

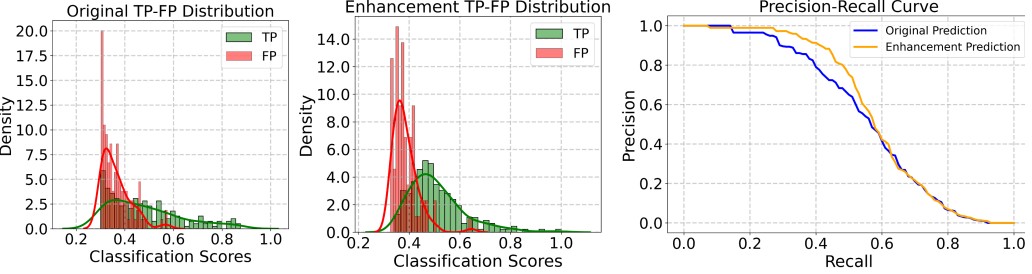

Figure 6: Analysis of Memory Enhancement.

**Analysis of Memory Enhancement.** We observe that Memory Enhancement remains effective even when only classification scores are modified without changing the predicted labels. Therefore, we conduct a comprehensive analysis of how Memory Enhancement improves the original predictions. In Fig. 6, we plot the True Positive (TP) and False Positive (FP) distributions of the original predictions and enhancement predictions for the category 'bicycle' on the Gaussian noise corruption of Pascal-C. It can be seen that Memory Enhancement optimizes the ordering of TP and FP predictions, ensuring that TP predictions are ranked higher than FP ones, thereby improving detection performance. The Precision-Recall curve for the 'bicycle' category further corroborates this result.

## 5 Conclusion

In this paper, we propose the first foundation model-powered test-time adaptive object detection method, which requires no source data while overcoming traditional closed-set limitations. The pro-

posed method employs a parameter-efficient Multi-modal Prompt-based Mean-Teacher framework for adaptation, incorporating a Test-time Warm-start strategy to preserve the teacher model's performance. Moreover, we introduce an Instance Dynamic Memory module, along with Memory Enhancement and Memory Hallucination strategies, to effectively leverage high-quality pseudo-labels from the test stream. Extensive evaluations on two benchmarks demonstrate that our method outperforms the state-of-the-art TTAOD approaches, with successful adaptation to arbitrary cross-domain and cross-category target data.

## Acknowledgment

This work is partly supported by the National Key Research and Development Plan (2024YFB3309302), the National Natural Science Foundation of China (62506111), the Postdoctoral Fellowship Program of CPSF (GZC20251098), the Research Program of State Key Laboratory of Complex and Critical Software Environment, the Anhui Postdoctoral Scientific Research Program Foundation (2025C1134) and the Fundamental Research Funds for the Central Universities.

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

# A  Appendix

## A.1  More Implementation Details

**Hyperparameters on the Cross-dataset Benchmark.** The cross-dataset benchmark is used to evaluate the detector's adaptability across both domains and categories, and consists of 13 object detection datasets: Ae (Aerial Maritime Drone), Aq (Aquarium), Co (Cottontail Rabbits), Eg (Egohands), Mu (Mushrooms), Pa (Packages), Pv (Pascal VOC), Pi (Pistols), Po (Pothole), Ra (Raccoon), Sh (Shellfish), Th (Thermal Dogs and People), Ve (Vehicles). Table 5 presents the detailed statistics of these datasets.

For the cross-dataset benchmark, we set $th_{pl}$ and $th_{me}$ to 0.3. The number $m$ of visual prompts is set to 5, and the maximum capacity $|Q_c|_{max}$ of IDM is set to 3. We use $\alpha = 1.0$ and $\beta = 5.0$ across all datasets. All other hyperparameters follows the settings in the main paper.

**Data Augmentation.** The strong augmentation consists of random resizing and one of color space transformation, selected from the following: ColorTransform, AutoContrast, Equalize, Sharpness, Posterize, Solarize, Color, Contrast, and Brightness. Additionally, it integrates RandErase, which randomly erases several patches (fewer than 5) with fixed pixel values at arbitrary locations to simulate occlusions. The weak augmentation consists solely of random resizing, enabling the teacher model to generate more reliable pseudo-labels for the student model.

Table 5: Datasets statistics of the cross-dataset benchmark.

| Dataset | Classes | Test Size |
|---|---|---|
| Ae | 5 | 15 |
| Aq | 7 | 127 |
| Co | 1 | 19 |
| Eg | 1 | 480 |
| Mu | 2 | 5 |
| Pa | 1 | 4 |
| Pv | 20 | 3,422 |
| Pi | 1 | 297 |
| Po | 1 | 133 |
| Ra | 1 | 29 |
| Sh | 3 | 116 |
| Th | 2 | 41 |
| Ve | 5 | 250 |

## A.2  More Experimental Results

Table 6: Full results about the ablation study of each component on Pascal-C.

| Methods | MPMT | | | IDM | | Noise | | | Blur | | | | Weather | | | | Digital | | | | Avg |
|---|---|---|---|---|---|---|---|---|---|---|---|---|---|---|---|---|---|---|---|---|---|
| | TPT | VPT | TTWS | ME | MH | Gauss | Shot | Impul | Defoc | Glass | Motn | Zoom | Snow | Frost | Fog | Brit | Contr | Elast | Pixel | Jpeg | |
| Direct Test | | | | | | 31.8 | 38.6 | 35.7 | 43.3 | 20.1 | 36.4 | 33.1 | 59.2 | 64.8 | 75.9 | 75.7 | 54.8 | 42.1 | 10.3 | 49.7 | 44.8 |
| Mean-Teacher | | | | | | 42.7 | 48.1 | 46.6 | 48.4 | 30.0 | 45.6 | 37.1 | 64.6 | 67.1 | 75.5 | 75.8 | 64.4 | 49.3 | 19.4 | 58.3 | 51.5 |
| (1) | ✓ | | | | | 32.7 | 39.2 | 36.5 | 43.3 | 22.4 | 37.2 | 33.3 | 60.5 | 65.0 | 75.5 | 75.5 | 55.2 | 44.2 | 9.9 | 50.7 | 45.4 |
| (2) | | ✓ | | | | 0.1 | 45.9 | 36.4 | 37.1 | 13.6 | 41.7 | 31.4 | 63.4 | 65.7 | 68.5 | 75.3 | 37.6 | 43.3 | 0.1 | 61.4 | 41.4 |
| (3) | | ✓ | ✓ | | | 45.4 | 50.0 | 49.3 | 46.1 | 32.0 | 44.5 | 36.3 | 64.0 | 66.8 | 76.1 | 76.3 | 64.2 | 51.5 | 37.6 | 61.6 | 53.4 |
| (4) | ✓ | ✓ | ✓ | | | 45.5 | 50.1 | 50.1 | 46.4 | 32.7 | 44.9 | 37.3 | 64.9 | 66.8 | 75.9 | 76.2 | 64.3 | 52.0 | 39.2 | 62.2 | 53.9 |
| (5) | | | | | ✓ | 33.2 | 39.6 | 37.2 | 44.6 | 20.6 | 37.9 | 34.2 | 61.1 | 66.4 | 77.5 | 77.0 | 55.6 | 43.5 | 10.7 | 52.0 | 46.1 |
| (6) | ✓ | ✓ | ✓ | | ✓ | 45.9 | 50.9 | 50.2 | 47.0 | 36.6 | 46.7 | 38.4 | 65.9 | 67.3 | 75.9 | 76.5 | 65.4 | 52.9 | 41.9 | 62.1 | 54.9 |
| Ours | ✓ | ✓ | ✓ | ✓ | ✓ | 46.9 | 52.0 | 51.9 | 47.9 | 38.6 | 48.6 | 39.4 | 66.7 | 68.6 | 77.7 | 77.8 | 66.5 | 54.2 | 42.8 | 63.7 | 56.2 |

**Full Results about the Ablation Study for Each Component.** In Table 6, we report the contribution of each component across all 15 corruption types on Pascal-C, as well as the average AP50. The proposed components achieve consistent improvements across nearly all types of corruption. We observe that using only Visual Prompt Tuning can lead to catastrophic failure during TTA on certain corruptions (*e.g.* Gaussian noise, Pixelate). Our proposed Test-time Warm-start strategy effectively addresses this issue.

Table 7: Comparison of the runtime cost and average AP50.

| Methods | Tuned Params↓ (M) | Latency↓ (ms/img) | Memory↓ (GB) | Avg↑ (%) |
|---|---|---|---|---|
| Full Fine-tuning | 164.964 | 635.0 | 20.9 | 51.5 |
| TPT | 0.037 | 532.9 | 16.6 | 45.4 |
| VPT | 0.042 | 562.7 | 17.6 | 53.4 |
| MPMT | 0.079 | 582.9 | 18.0 | 53.9 |

**Runtime Cost.** In Table 7, we present the number of learnable parameters, per-image test-time adaptation latency, peak GPU memory footprint, and average AP50 on Pascal-C, measured using an RTX 3090 GPU. Compared to Full Fine-tuning, our proposed Multi-modal Prompt-based Mean-Teacher

framework requires only 0.05% of the learnable parameters, enabling us to store corresponding multi-modal prompts for each target domain while sharing a single copy of the pre-trained Grounding DINO weights. This advantage also facilitates easy extension to Continual Test-time Adaptive Object Detection (CTTAOD). In addition, our method demonstrates benefits in latency and peak GPU memory footprint compared to Full Fine-tuning, while achieving a significant improvement in Average AP50.

**About the Threshold of Memory Enhancement.** Since Grounding DINO generates 300 predictions per image, applying Memory Enhancement to all predictions would significantly reduce inference speed. As shown in Fig. 7, we select the threshold $th_{me}$ under Gaussian noise corruption on Pascal-C. When $th_{me}$ is below 0.2, the latency (blue solid line) increases notably compared to direct testing (blue dashed line). Moreover, we observe that when $th_{me}$ is below 0.2, Memory Enhancement leads to a performance decline (green solid line) compared to direct testing (green dashed line). This is because, at this threshold, there are a large number of noisy predictions, and refining noisy predictions causes greater confusion in the final results. Considering both latency and detection performance, we set it to 0.3.

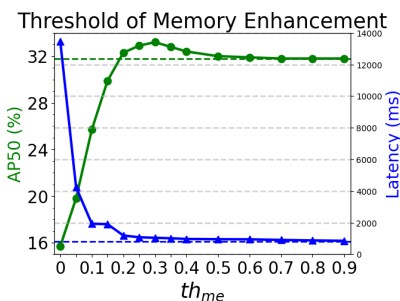

Figure 7: Analysis of the Memory Enhancement threshold.

**Sensitivity Analysis on the Weighting Factor and Sharpness Ratio.** We conduct a sensitivity analysis of the hyperparameters $\alpha$ and $\beta$ on the Gaussian noise corruption of Pascal-C. As shown in Table 8, we achieve the best AP50 when setting $\alpha$ to 5.0 and $\beta$ to 5.0.

Table 8: Sensitivity analysis of $\alpha$ and $\beta$.

| $\alpha$ | 2.0 | 3.0 | 4.0 | **5.0** | 6.0 | 7.0 |
|---|---|---|---|---|---|---|
| | 32.5 | 32.8 | 33.0 | **33.2** | 32.8 | 32.7 |
| $\beta$ | 0.5 | 1.0 | 3.0 | **5.0** | 7.0 | 9.0 |
| | 32.6 | 32.7 | 33.0 | **33.2** | 32.4 | 32.0 |

Table 9: Comparison of the inference cost and average mAP on COCO-C.

| Detectors | Methods | Source Data | Latency (ms/img) | Avg |
|---|---|---|---|---|
| Faster RCNN (Res-50) | Direct Test | Pre-train | 54.2 | 15.9 |
| | Mean-Teacher [30] | Pre-train | 296.1 | 18.6 |
| | STFAR [2] | Pre-train + Statistical Characteristics | 327.3 | 21.7 |
| | CTTAOD [31] | Pre-train + Statistical Characteristics | 143.9 | 22.3 |
| | CTTAOD-Skip [31] | Pre-train + Statistical Characteristics | 84.9 | 21.7 |
| Grounding DINO (Swin-T) | Direct Test | × | 213.1 | 20.6 |
| | Mean-Teacher [30] | × | 635.0 | 24.1 |
| | Ours | × | 693.8 | 26.0 |

**Detailed Analysis of Inference Costs.** TTA not only requires inference but also involves model adaptation based on the current test data. This process inevitably introduces additional latency compared to direct testing. As illustrated in Table 9, although previous approaches demonstrate faster inference speeds, they require pre-training weights for each source domain and extract statistical characteristics from the source data—both of which incur significant time costs. Compared to the strong baseline Mean-Teacher, we introduce only minimal additional inference time (693.8 ms/img versus 635.0 ms/img), and achieve nearly a 2% performance improvement on COCO-C.

On the other hand, some studies have begun to specifically focus on improving efficiency. For example, CTTAOD-Skip [31] improves average inference speed by skipping some test samples, and Efficient TTAOD [29] accelerates inference through pruning. We attempted to integrate a relatively simple Skip strategy into our method, which reduces the latency to 387.7 ms/img while maintaining comparable performance with a minimal performance drop.

### A.3 Visualization

**Examples of Memory Hallucination.** Fig. 8 shows examples of Memory Hallucination. It can be observed that Memory Hallucination effectively leverages high-quality instances from the Instance

Dynamic Memory, as well as negative images without available pseudo-labels, to generate diverse positive samples.

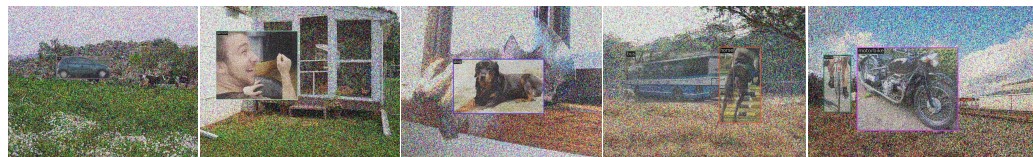

Figure 8: Examples of Memory Hallucination.

**Examples of Memory Enhancement.** In Fig. 9, we present the changes in predictions before and after applying Memory Enhancement. Red boxes indicate false positive (FP) samples, while green boxes represent true positive (TP) samples. In the first row, the original classification scores of the two FP samples are higher than those of the two TP samples. After applying Memory Enhancement, as shown in the second row, the TP samples receive significant enhancement, resulting in their classification scores surpassing those of the two FP samples. The same trend is observed in the third and fourth rows. Memory Enhancement improves the recall of high-confidence predictions, enabling the detector to produce predictions that include more true positives.

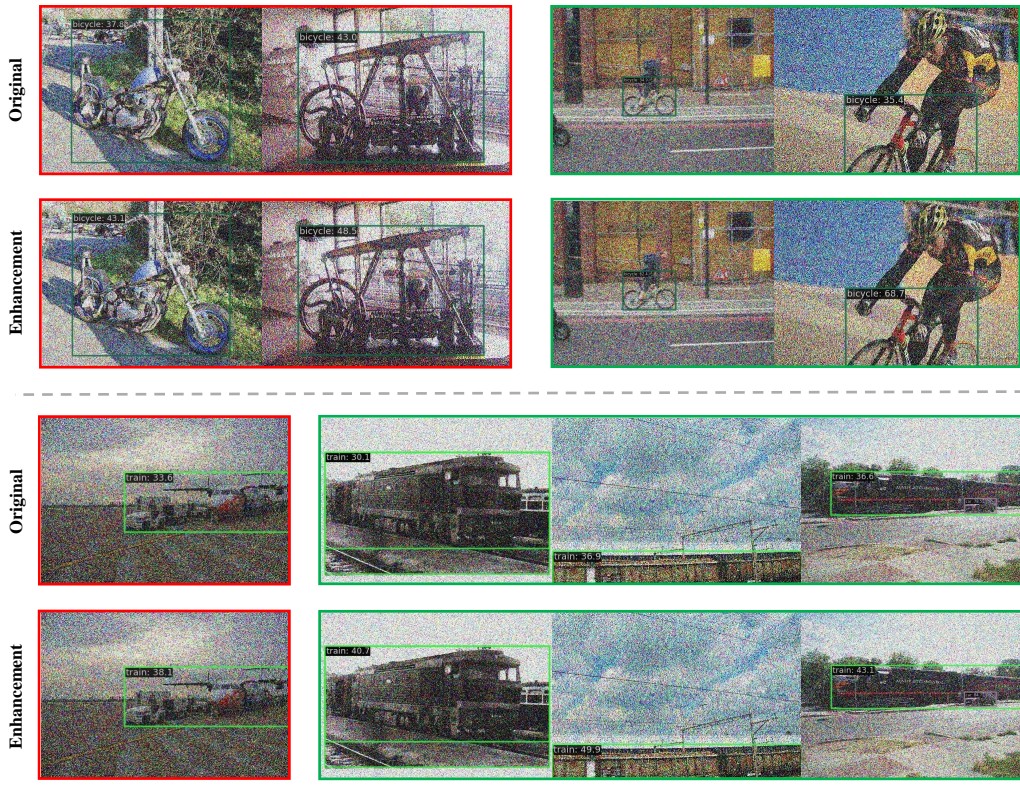

Figure 9: Examples of Memory Enhancement.

**Detection Results.** We visualize detection results with all comparative methods based on Grounding DINO. Fig. 10 shows the detection results on Pascal-C under Gaussian noise corruption. Our method alleviates misclassifications, as seen in the first and second rows, missed detections, as shown in the third row, and false positives, as illustrated in the fourth row. We also provide visualizations under different corruption types (Fig. 11) and across various corruption types on different datasets ( Fig. 12), further demonstrating the effectiveness of our method.

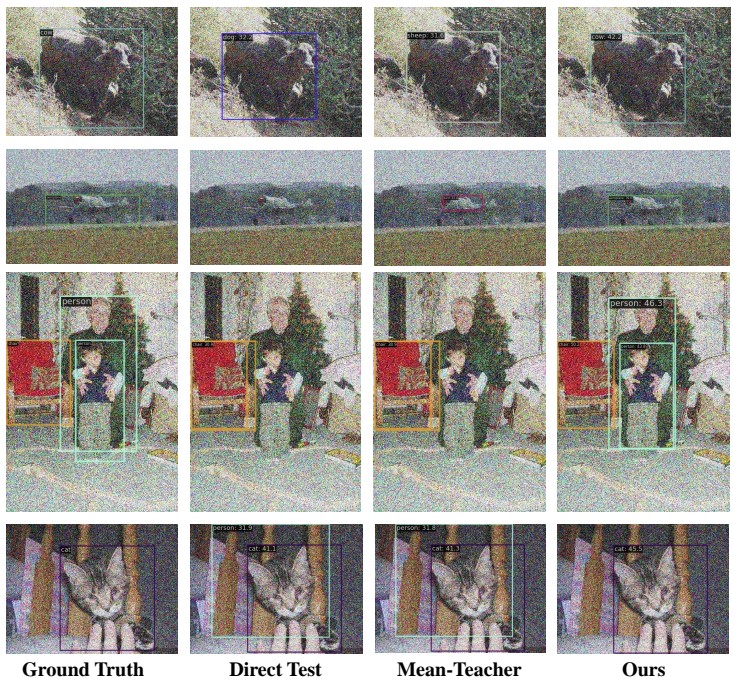

Figure 10: Detection results under Gaussian noise corruption on Pascal-C.

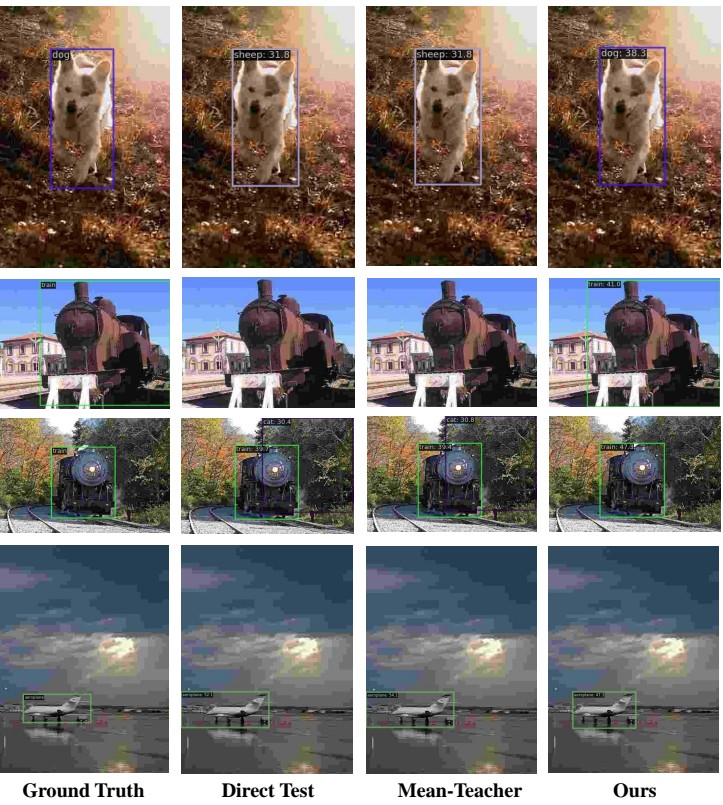

Figure 11: Detection results under JPEG compression corruption on Pascal-C.

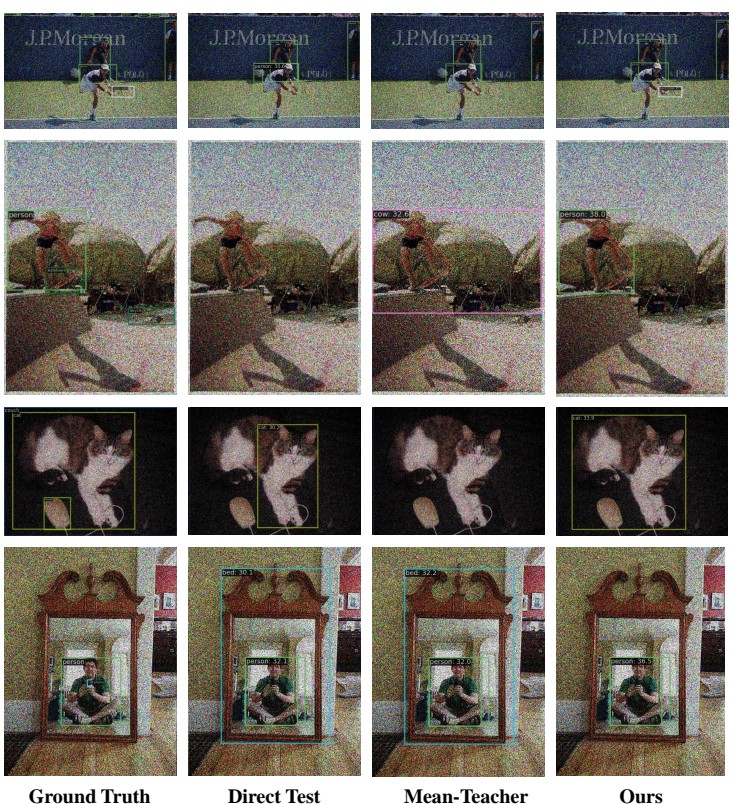

| **Ground Truth** | **Direct Test** | **Mean-Teacher** | **Ours** |

Figure 12: Detection results under Shot noise corruption on COCO-C.

