# OpenReview forum: "Test-Time Adaptive Object Detection with Foundation Model"
_NeurIPS.cc/2025/Conference — NeurIPS 2025 poster_

### Official Review · Reviewer_yLfD · 2025-07-01

**Clarity:** 3
**Significance:** 3
**Originality:** 3
**Rating:** 4
**Confidence:** 4

**Summary:**

This paper proposes a multi-modal prompt based mean-teacher (MPMT) framework for the problem of test-time adaptive object detection (TTAOD), and utilizes a vision-language detector for self-training during TTA. The paper introduces an Instance DynamicMemory (IDM) module, and it is maintained for each category during test time. Based on IDM, they propose two schemes, namely, memory reinforcement (MR) and memory hallucination (MH), to utilize good quality pseudo-labels from previous examples. Experiments on cross-corruption and cross-dataset benchmarks demonstrate improved performance for the proposed approach compared to other approaches.

**Questions:**

1. How will the proposed approach perform when it encounters data from entirely new categories or when there is a significant distribution shift not represented in the pre-training data?
2. Can the IDM module handle distribution drift in the continual test-time settings? Will the error accumulation due to noisy pseudo-labels cause a significant performance degradation in a continual scenario?
3. What is the runtime latency compared to the direct test? Such comparisons of wall clock times will be helpful in understanding the computational overhead during inference.
4. How sensitive are the results to the other hyperparameters, such as $th_{pl}$ and memory capacity? Whether the hyperparameter tuning is performed using the validation split?

**Ethical Concerns:**

["NO or VERY MINOR ethics concerns only"]

**Final Justification:**

The authors have responded to most of the queries.

As already discussed in "W1: About fair comparison", a fair comparison is challenging, and a discussion and more thought about it are required.

I will maintain a positive score.

**Limitations:**

* Some other limitations, such as scaling to drastic distribution shifts, may be included.
* The trade-offs between accuracy and inference speed can be discussed.

**Quality:**

3

**Strengths And Weaknesses:**

**Strengths**
* This work utilizes a vision language foundation models (VLMs) for doing TTAOD.
* Use of VLMs allows overcoming the closed-set limitation.
* MPMT is a nice way to adapt the large foundation model in a parameter efficient manner.
* Ablation study shows the contributions of different components of the model

**Weaknesses**
* The paper's claim of outperforming state-of-the-art methods is based on an unfair comparison. The proposed approach uses a powerful VLM (Grounding DINO, SWIN_T) as it's backbone, whereas, the compared prior works uses standard or less powerful architectures like Faster-RCNN (ResNet-50/101). Thus, it is impossible to isolate the actual improvement of the proposed adaptation approach compared to prior state-of-the-art approaches.
* The quality of the pseudo-labels depends on the threshold $th_{pl}$. More exploration for this hyperparameter is lacking.
* During test-time, the computational pipeline will have a forward pass involving a large detector, conditional feature extraction with another large model (DINOv2) and memory updates. This will limit its applicability in truly real-time and resource-constrained settings.

**Typos**
* Line 117: "Groudning" -> "Grounding"
* Figure 2: "Updata"

---

> ### Author Rebuttal · Authors · 2025-07-31
>
> **W1: About fair comparison.**
>
> Previous methods rely on source domain data for fine-tuning or statistical computation, whereas our method has no access to source domain data. And considering the differences in the adopted detectors, achieving a completely fair comparison is challenging. However, we have made efforts to ensure a fair comparison with previous methods. For example, in Table 2, we compare with CTTAOD based on Faster R-CNN (Swin-T).  Its results are even worse than using ResNet-50/101 as backbones, because ResNet-50/101 use SoftTeacher weights pretrained on the COCO training set with extensive data augmentation.
>
> We also conduct thorough comparisons with the strong baseline Mean-Teacher using Grounding DINO, which demonstrate the performance improvements achieved by our method.
>
> **W2: Ablation on the threshold $th_{pl}$.**
>
> Thank you for pointing out this issue.  We select thethreshold $th_{pl}$ under Gaussian noise corruption on Pascal-C. When $th_{pl}$ is too high, only a small number of pseudo-labels are available for test-time adaptation, which hinders the model from effectively adapting to the target domain. Conversely, if $th_{pl}$ is set too low, noisy pseudo-labels may be introduced, which can also degrade the model's performance. Therefore, we set it to 0.3.
>
>
> |  $th_{pl}$  | 0.1  | 0.2  | 0.3  |0.4  |0.5  |
> | :-------: | :--: | :--: | :--: |:--: |:--: |
> | VOC-Gauss | 45.0 | 46.0 | **46.9** |45.6|41.2|
>
> **W3: About the applicability in real-time and resource-constrained settings.**
>
> The latency of our method compared to directly testing with a vision-language detector is 693.8 ms/img vs. 213.1 ms/img. However, it brings significant performance improvements, with gains of +11.4\% AP50 on Pascal-C and +5.4\% mAP on COCO-C. For more analysis of inference speed and storage costs, please refer to our response to Reviewer 4Xpg W1.
>
> **Typos:** Thank you, we will revise the typos.
>
> **Q1: The performance under significant distribution shifts or unseen categories.**
>
> For Pascal-C and COCO-C, we follow previous works and use the most challenging corruption level for evaluation.  Additionally, ODinW-13 contains 13 datasets with significant domain shifts (e.g. remote sensing, underwater, etc.).  These benchmarks ensure significant distribution shifts from the pre-training data. On the other hand, since the pre-training data of Grounding DINO includes grounding data, it is difficult to determine whether the categories in OdinW-13 are entirely novel. Our key contribution, by contrast, lies in introducing a vision-language detector to overcome the closed-set limitation. The experimental results in Table 1, Table 2, and Fig. 4 demonstrate that our method performs well under significant distribution shifts and in open-set scenarios.
>
> **Q2: About the IDM module for distribution drift in continual test-time settings and the error accumulation from noisy pseudo-labels in continual scenario.**
>
> Our method focuses on TTAOD, and continual test-time setting is beyond the scope of our paper. Theoretically, the IDM module can be extended to handle distribution drift in a continual test-time scenario. Like CTTAOD [1], we can detect sudden distribution shifts to identify the arrival of a new target domain. Then, we clear the triplets in the IDM module and re-maintains them on the new target domain to handle distribution drift. As for error accumulation, upon detecting the arrival of a new target domain, we can reinitialize new text-visual prompts in our MPMT to alleviate this issue.
>
> [1]Yoo J, Lee D, Chung I, et al. What how and when should object detectors update in continually changing test domains?[C]. In CVPR, 2024.
>
> **Q3: About the runtime latency compared to direct testing.**
>
> Please refer to our detailed response to W3.
>
> **Q4: About the hyperparameters $th_{pl}$ and memory capacity.**
>
> The ablation study of the threshold $th_{pl}$ is provided in W2, and the analysis of IDM’s memory capacity is presented in Fig. 5 and Section 4.4 of the paper. Regarding hyperparameter tuning, we strictly followed prior work by selecting hyperparameters under a single type of corruption (Gaussian noise in our paper).
>
> **Limitations:** Please refer to Q1 and W3. Thank you for the reminder. We will include a discussion on the detection performance and inference speed of our method in the limitations section.

---

> > ### Comment · Reviewer_yLfD · 2025-08-01
> > **Acknowledgement and Further Queries**
> >
> > Thanks to the authors for responding to some of my queries.
> >
> > **W1: About fair comparison.**
> >
> > As mentioned in your response, a fair comparison is indeed challenging. However, this point is not discussed in the paper, neither in the main paper nor in a dedicated "Limitations" section.
> >
> > **W2 & Q2: Ablation on the threshold $th_{pl}$**
> >
> > Is the same threshold used for the COCO-C dataset? What about the sensitivity of this threshold value with respect to different architectures?
> >
> > **W3: About the applicability in real-time and resource-constrained settings.**
> >
> > Latency of 693.8 ms/img vs. 213.1 ms/img is high, i.e., ~3.25 times slower.
> > Even though the performance improvement is appealing, if we are have higher number of images (say, multiple frames from a video), the latency will be a bottleneck.
> >
> > I will keep my score unchanged for now.
> >
> > Thank you.

---

> > > ### Author Response · Authors · 2025-08-03
> > >
> > > Thanks for your timely responses.
> > >
> > > **W1: About fair comparison.**
> > >
> > > Thank you for your suggestion. We will add a discussion on fair comparisons to the Experiments Section and Limitations Section in the main paper.
> > >
> > > **W2 & Q2: Ablation on the threshold $th_{pl}$**
> > >
> > > The same threshold is also used on the COCO-C dataset. We conducted a sensitivity analysis across different corruption types in COCO-C, as shown in the table below.
> > >
> > > | $th_{pl}$  | 0.1  | 0.2  |   0.3    | 0.4  | 0.5  |
> > > | :--------: | :--: | :--: | :------: | :--: | :--: |
> > > | COCO-Gauss | 19.0 | 19.9 | **20.2** | 19.0 | 16.3 |
> > > | COCO-Defoc | 17.4 | 17.5 | **17.8** | 17.3 | 16.4 |
> > >
> > > We also conducted a sensitivity analysis of the threshold $th_{pl}$ across different detector architectures under Gaussian noise corruption on Pascal-C.
> > >
> > > | $th_{pl}$  | 0.1  | 0.2  | 0.3  | 0.4  | 0.5  |
> > > | :--------: | :--: | :--: | :--: | :--: | :--: |
> > > | Swin-B | 63.5 | 63.6 | **64.9** | 63.9 | 63.2 |
> > > | Swin-L | 74.3 | 74.7 | **76.2** | 75.8 | 74.2 |
> > >
> > > The results show that the threshold is insensitive to both datasets and detector architectures.
> > >
> > > **W3: About the applicability in real-time and resource-constrained settings.**
> > >
> > > TTA not only requires inference but also needs to perform adaptation based on the current test data. Therefore, it inevitably introduces additional latency compared to direct testing.  For example, existing methods such as STFAR, CTTAOD, and the Mean-Teacher baseline also increase latency by \~6.04, \~2.65 and \~5.46 times respectively, compared to their direct test versions. The latency increasing also exists in TTA methods for image classification, such as TPT (\~12h) and diffTPT (\~34h), compared to direct testing with CLIP (\~12min) on ImageNet.
> > >
> > > A proper comparison would be against the strong baseline Mean-Teacher, where we introduce only minimal additional inference time (693.8 ms/img vs. 635.0 ms/img) and achieves nearly a 2% performance improvement on COCO-C.
> > >
> > > Overall, all existing TTA methods do increase latency to certain extent. On the other hand, some studies have  begun to specifically focus on improving efficiency. For example, CTTAOD-Skip improves average inference speed by skipping some test samples, and Efficient TTAOD accelerates inference through pruning.  Although this is not our main contribution, we still attempt to incorporate existing methods to optimize our inference speed. Due to time limits, we attempted to integrate a relatively simple Skip strategy into our method. The latency decreases to 387.7 ms/img, while the performance drops only 0.6%  AP50. Theoretically, pruning can also be applied to our approach to improve computational efficiency, which we plan to explore in the future.
> > >
> > > |   Methods   | Latency (ms/img) | VOC-Gauss |
> > > | :---------: | :--------: | :------: |
> > > | Direct Test |    213.1    |    31.8    |
> > > |     Mean-Teacher     |    635.0    |   42.7   |
> > > | Ours |    693.8    |    46.9    |
> > > |     Ours-Skip     |    387.7    |   46.3   |
> > >
> > > Finally, we would like to recall the advantage of our method. Although our method increases some latency compared to traditional Faster R-CNN based approaches, the core contribution of introducing VLMs also represents a significant breakthrough, as it overcomes the closed-set limitation and entirely eliminates dependence on source domain data.

---

> > > ### Author Response · Authors · 2025-08-07
> > >
> > > Dear Reviewer,
> > >
> > > As the reviewer-author discussion phase nears its end, we kindly ask for your prompt feedback on our rebuttal. We are eager to know whether our efforts have successfully addressed your concerns. If all your concerns have been resolved, we would sincerely appreciate it if you could consider revising the final rating.
> > >
> > > Best regards,
> > >
> > > Authors.

---

> > > > ### Comment · Reviewer_yLfD · 2025-08-07
> > > > **Acknowledging further response from the authors**
> > > >
> > > > Thanks for responding to my further queries.
> > > >
> > > > I will consider all these aspects before submitting my final score after the AC-reviewer discussion phase.
> > > >
> > > > I will keep my positive score unchanged now.

---

### Official Review · Reviewer_4Xpg · 2025-07-03

**Clarity:** 2
**Significance:** 2
**Originality:** 2
**Rating:** 4
**Confidence:** 4

**Summary:**

The authors introduce a novel test-time adaptive object detection method powered by vision-language foundation models GDINO, which eliminates the need for source data and overcomes the traditional closed-set limitations. The proposed method adopts a Multi-modal Prompt-based Mean-Teacher framework, where learnable vectors are added to the token embeddings of both the text and image branches to enable effective prompt tuning. In addition, Test-time Warm-start strategy is employed to preserve the model’s representation capability during adaptation. The Instance Dynamic Memory IDM module stores high-quality pseudo-labels from previous test samples and supports two enhancement strategies: Memory Reinforcement and Memory Hallucination to improve prediction accuracy and generate positive samples for challenging images. Experiments conducted on multiple datasets demonstrate that the proposed method achieves state-of-the-art performance. Comprehensive ablation studies further highlight the individual contributions of each component in the framework.

**Questions:**

1. The change in mAP of the proposed method after each epoch should be reported.
2. Present the ablation results showing the impact of using TPT, VPT, TTWS, and MR.
3. Provide the actual latency of the proposed method compared to previous approaches.
4. Include comparisons with recently proposed state-of-the-art (SOTA) methods.

Typos etc:
L117: use “Grounding” instead of “Groudning”; l289: “Tabel”; l521:”poistive”

**Ethical Concerns:**

["NO or VERY MINOR ethics concerns only"]

**Final Justification:**

The authors have clearly and thoroughly addressed my questions in the rebuttal, particularly clarifying the issues related to inference speed and storage cost. Therefore, I have changed my rating for this paper to 4.

**Limitations:**

No.

**Paper Formatting Concerns:**

No.

**Quality:**

3

**Strengths And Weaknesses:**

Strengths:
1. The proposed method leverages a vision-language foundation model (Grounding DINO) for test-time adaptive object detection without requiring any access to source data, and is capable of adapting to novel label sets beyond those seen during initial training (open-vocabulary adaptation)
2. The multi-modal prompt-based Mean-Teacher framework, which incorporates both language and visual prompt tuning, enables the model to effectively adapt to target data by fine-tuning only a very small number of parameters, thereby reducing training time.

Weaknesses:
1. The use of Grounding DINO (Swin-T) significantly slows down the model’s inference speed and increases storage costs when Instance Dynamic Memory is employed. Please provide a detailed explanation for these limitations.
2. The approach of adding a learnable vector to the image and text tokens may lead to representation collapse in the absence of source data. (see questions)

---

> ### Author Rebuttal · Authors · 2025-07-31
>
> **W1: About inference speed and storage costs.**
>
> Thank you for pointing out the potential limitation. As shown in the table below, compared to the Mean-Teacher baseline using Grounding DINO, our approach achieves nearly a 2\% performance improvement with only minimal additional inference time.  And our method overcomes the closed-set limitation and enables us to completely eliminate reliance on source domain data. Although previous approaches have faster inference speeds, they require pre-training weights for each source domain data and extract statistical characteristics from the source data—both of which also incur significant time costs. For example, pre-training takes about 2 hours for VOC and 15 hours for COCO on 8 V100 GPUs.
>
> In the following table, CTTAOD-Skip* demonstrates excellent latency performance, primarily because it skips adaptation for some test samples based on the distribution gap. Recent works, such as Efficient TTAOD [1], also specifically focus on the efficiency issue. These techniques can also be integrated into our method to further improve inference speed.
>
> |   Detectors    |   Methods    |Use Source Data | Latency (ms/img) | COCO-C (Avg) |
> | :----------:| :----------: | :-----: | :--: | :--: |
> |Faster RCNN| Direct Test  |Pre-train|   54.2   |15.9|
> |(ResNet-50)| Mean-Teacher |Pre-train|   296.1  |18.6|
> ||    STFAR    | Pre-train + Statistical Characteristics |  327.3  |21.7|
> ||    CTTAOD    | Pre-train + Statistical Characteristics |  143.9  |22.3|
> || CTTAOD-Skip* | Pre-train + Statistical Characteristics |  84.9    |21.7|
> |Grounding DINO| Direct Test  | No |  213.1  |20.6|
> |(Swin-T)| Mean-Teacher |No|   635.0  |24.1|
> ||     Ours    | No |  693.8   |26.0|
>
> Employing IDM does introduce an increase in storage cost, primarily due to loading DINOv2-L, which requires approximately 1 GB. The triplets maintained by IDM itself require only about 50 MB. In real-world deployments, techniques like quantization and pruning can be considered to reduce storage costs.
>
> Thank you again. We will provide a detailed discussion of these limitations in the paper.
>
> [1]Wang K, Fu X, Lu X, et al. Efficient Test-time Adaptive Object Detection via Sensitivity-Guided Pruning[C]. In CVPR, 2025.
>
> **W2: Preventing representation collapse.**
>
> This is a key issue that we have also noticed, and we have already discussed it in Section 3.3. Since the text prompt is added to the language branch in the form of a residual (Eq. 1), representation collapse can be avoided by initializing it with all zeros. For the visual prompts, the proposed TTWS is specifically designed to address this issue, as shown in Fig. 3.
>
> We include results for directly testing with added learnable vectors in the following table. Using randomly initialized learnable vectors indeed leads to representation collapse, but this issue can be addressed by incorporating the proposed TTWS. Furthermore, representation collapse can cause catastrophic collapse of the detector during test-time adaptation. Our proposed TTWS effectively addresses this issue, as demonstrated in Tables 3 and 6 of the paper.
>
> |Methods|VOC-Gauss|VOC-Shot|VOC-Impul|
> |:--------:|:--------:|:--------:|:--------:|
> |Direct Test|31.8|38.6|35.7|
> |+Random Init|17.2|24.1|18.4|
> |+TTWS|33.1|39.1|36.4|
>
> **Q1: The change in mAP after each epoch.**
>
> First, there may be a misunderstanding. In TTA, we perform adaptation with only one pass over the test data, and the model learns while making predictions. The evaluation is conducted using the accumulated prediction results. We will emphasize this point in the paper. To more clearly demonstrate the evolution of mAP, we evaluated our method at different iterations, and the results show that as the number of iterations increases, the model's performance gradually improves, indicating a more thorough adaptation to the target domain.
>
> |Iters|500|1000|1238 (End)|
> |:--------:|:--------:|:--------:|:--------:|
> |VOC-Gauss|46.5|48.6|50.1|
>
> **Q2: About the ablation results.**
>
> We have already investigate the impact of using TPT, VPT, TTWS, and MR in Table 3, and provide detailed results in Table 6. This might have been overlooked.
>
> **Q3: About the latency.**
>
> Please refer to our response to W1.
>
> **Q4: Comparisons with recently proposed SOTA methods.**
>
> We have compared our method with recent SOTA approaches, including STFAR, CTTAOD, and the strong baseline Mean-Teacher. We also noticed a recent work, Efficient TTAOD*[1], which primarily focuses on the efficiency issue in Continual Test-time Object Detection. Since its code has not been released, we are only able to compare with it on its first target domain of the first run. Additionally, we compare our method with several TTA approaches designed for image classification [2,3,4]. As illustrated in the following table, our method achieves SOTA performance.
>
> |   Methods   | Cityscapes | ACDC-snow  |
> | :---------: | :--------: | :------: |
> | Direct Test |    -    |    21.8    |
> |     Efficient TTAOD*     |    -    |   23.4   |
> | Direct Test |    30.5    |    33.9    |
> |     TPT     |    30.9    |   34.4    |
> |   DiffTPT   |    31.2    |   34.6   |
> |   HisTPT    |    31.9    |   35.6   |
> | Direct Test |    50.2    |    52.5    |
> |    Ours     |  **55.2**  | **54.1** |
>
> [2] Shu M, Nie W, Huang D A, et al. Test-time prompt tuning for zero-shot generalization in vision-language models[J]. In NeurIPS, 2022.
>
> [3] Feng C M, Yu K, Liu Y, et al. Diverse data augmentation with diffusions for effective test-time prompt tuning[C]. In  ICCV, 2023.
>
> [4] Zhang J, Huang J, Zhang X, et al. Historical test-time prompt tuning for vision foundation models[J]. In NeurIPS, 2024.
>
> **Typos:** Thank you. We have carefully proofread the paper and corrected the typos.

---

> > ### Comment · Reviewer_4Xpg · 2025-08-05
> >
> > Thank you for your very detailed rebuttal. You have addressed all of my concerns. Good luck to you.

---

> > > ### Author Response · Authors · 2025-08-05
> > >
> > > Thank you for your constructive comments and acknowledgment of our work. We are delighted that we have addressed all your concerns. If there are no further concerns, we kindly hope you will consider updating the final rating.
> > >
> > > Best regards,
> > >
> > > Authors.

---

### Official Review · Reviewer_E5Mt · 2025-07-03

**Clarity:** 4
**Significance:** 4
**Originality:** 4
**Rating:** 5
**Confidence:** 5

**Summary:**

The paper proposes a novel test-time adaptive object detection approach based on the vision-language model. The proposed method employs a self-training framework that fine-tunes only the text and visual prompts. Specifically, the visual prompts are initialized using a test-time warm-start strategy to prevent performance degradation caused by inserting prompts into the visual branch. Additionally, the paper introduces an Instance Dynamic Memory module for storing high-quality pseudo labels, which are used to enhance original predictions and perform data augmentation for images without available pseudo labels. Experimental results demonstrate that the proposed method achieves superior performance across multiple benchmarks.

**Questions:**

See Weaknesses section above.

**Ethical Concerns:**

["NO or VERY MINOR ethics concerns only"]

**Final Justification:**

Thanks for the rebuttal comment. All of my concerns have been addressed. However, I still suggest that the authors supplement more details and analysis in the camera ready version. For example, as pointed out by yLfD02, the fair comparison and real-time inference are important for this task so the authors should add more detailed clarifications on them. Overall, I will keep my original rating.

**Quality:**

4

**Strengths And Weaknesses:**

**Strengths:**

1.	This paper is well-motivated, addresses a more practical problem in TTAOD, and is the first to introduce a vision-language model into this field.
2.	The proposed method is easy to understand and follow.
3.	The proposed method demonstrates superior performance compared to existing Faster R-CNN based TTAOD methods and the VLM based baseline.
4.	The experimental results on both cross-corruption and cross-dataset benchmarks are abundant, along with comprehensive ablation studies.


**Weaknesses:**

1.	The authors should compare their proposed multi-modal prompt tuning approach with other commonly used parameter-efficient fine-tuning methods, such as LoRA.
2.	The authors use DINOv2 features when maintaining the Instance Dynamic Memory module, it is suggested that the authors provide an explanation for this choice.
3.	The paper only conducts experiments based on Swin-T Grounding DINO. Would using larger model sizes bring further improvements?
4.	The names Memory Reinforcement and Memory Hallucination proposed in the paper may cause confusion with concepts from reinforcement learning and LLM’s hallucination. It is recommended that the authors consider revising these terms.

---

> ### Author Rebuttal · Authors · 2025-07-31
>
> **W1: Compare with other PEFT methods.**
>
> Thank you for your suggestion!  Prompt tuning has been widely used for fine-tuning VLMs and has achieved impressive results. Following your advice, we compared our method with two PEFT approaches, LoRA and Adapter, based on the implementations provided in [1]. Both LoRA and Adapter show improvements over direct testing, but our method consistently achieves superior performance.
>
> |Methods|VOC-Gauss|
> |:--------:|:--------:|
> |Direct Test|31.8|
> |Mean-Teacher|42.7|
> |LoRA|43.5|
> |Adapter|39.6|
> |Ours|**46.9**|
>
> [1] Yin D, Hu L, Li B, et al. 5\% $>$ 100\%: Breaking performance shackles of full fine-tuning on visual recognition tasks[C]. In CVPR, 2025.
>
> **W2: Reasons for adopting DINOv2 for IDM.**
>
> DINOv2 is pretrained on large curated data from diverse sources, enabling highly generalizable feature representations. To determine the optimal feature extractor for the IDM module, we conducted a comprehensive comparison between DINOv2 and CLIP across various model sizes. Experiments demonstrate DINOv2's superior performance over CLIP, with model scaling further enhancing MR. Considering computational efficiency, we use DINOv2-L to maintain the IDM module.
>
> |Methods|Params (M)|VOC-Gauss|
> |:--------:|:--------:|:--------:|
> |DINOv2-B/14|86|32.3|
> |DINOv2-L/14|300|33.2|
> |DINOv2-g/14|   1,100    |33.7|
> |CLIP-B/32|87|31.9|
> |CLIP-B/16|86|32.0|
> |CLIP-L/14|304|32.4|
>
> **W3: Performance with large backbones.**
>
> Considering computational cost, Swin-T is commonly used in tasks based on vision-language detectors [2]. Accordingly, we also adopt Swin-T for our experiments. In general, using larger models yields further performance improvements, which is also confirmed by the results shown in the table below. Moreover, our method consistently brings performance gains across Swin-T, Swin-B, and Swin-L.
>
> |Backbones|Methods|VOC-Gauss|
> |:--------:|:--------:|:--------:|
> |Swin-T|Direct Test|31.8|
> ||Ours|46.9|
> |Swin-B|Direct Test|57.2|
> ||Ours|64.9|
> |Swin-L|Direct Test|71.2|
> ||Ours|76.2|
>
> [2] Deng J, Zhang H, Ding K, et al. Zero-shot generalizable incremental learning for vision-language object detection[J]. In NeurIPS, 2024.
>
> **W4: Rename Memory Reinforcement and Memory Hallucination.**
>
> Thanks for your suggestion. We will revise "Memory Reinforcement" to "Memory Enhancement". As for "Memory Hallucination", in the context of few-shot learning [3], "hallucination" is commonly used to refer to the generation of new samples based on existing data. Therefore, we adopt this term. If you have a better suggestion, we would be glad to consider it.
>
> [3] Zhang W, Wang Y X. Hallucination improves few-shot object detection[C]. In CVPR, 2021.

---

> > ### Comment · Reviewer_E5Mt · 2025-08-05
> >
> > Thanks for the rebuttal comment. All of my concerns have been addressed. However, I still suggest that the authors supplement more details and analysis in the camera ready version. For example, as pointed out by yLfD02, the fair comparison and real-time inference are important for this task so the authors should add more detailed clarifications on them. Overall, I will keep my original rating.

---

> > > ### Author Response · Authors · 2025-08-05
> > >
> > > Thank you for acknowledging the core contributions of our work! We are glad that we have addressed all your concerns. We will include more detailed clarifications on fair comparison and real-time inference in the camera ready version.
> > >
> > > Finally, we sincerely appreciate your efforts to improve our work.

---

### Note · Authors · 2025-08-13

Dear AC and Reviewers,

We sincerely thank you for your efforts and insightful feedback to improve our work!

We are pleased that all reviewers consistently appreciate our work's novelty, effectiveness, and extensive experiments.

- This paper is well-motivated  (Reviewer *E5Mt*) and leverages a vision-language model for TTAOD (All Reviewers).
- The proposed method does not require any access to source data  (Reviewer *4Xpg*) and overcomes the closed-set limitation  (Reviewer *4Xpg* and *yLfD*).
- The MPMT enables the model to effectively adapt to target data (Reviewer *4Xpg* and *yLfD*).
- The proposed method demonstrates superior performance  (Reviewer *E5Mt*).
- The experiments are sufficent, and the ablation studies show the contributions of different components (Reviewer *E5Mt* and *yLfD*).

We are confident that we have thoroughly addressed all of the concerns raised by Reviewers *E5Mt* and *4Xpg*. Reviewer *yLfD* also acknowledged our responses and maintained a positive score.  We will incorporate these valuable suggestions into the revised version.

Thanks for your time and efforts again!

Best regards,

Authors.

---

### Decision · Program_Chairs · 2025-09-17

**Decision:**

Accept (poster)

**Comment:**

This paper introduces a framework for test-time adaptive object detection (TTAOD) by leveraging a vision-language foundation model. The ky contribution is a method that eliminates the need for source data and overcomes the traditional closed-set limitations of TTAOD, enabling adaptation to new domains and categories. This is achieved through a Multi-modal Prompt-based Mean-Teacher framework for parameter-efficient adaptation and an Instance Dynamic Memory module to ensure high-quality pseudo-labels. The reviewers unanimously recognized the work's novelty, strong empirical performance across multiple benchmarks, and thorough experimental validation. The primary weaknesses identified during the review process were the potential for an unfair comparison due to the use of a stronger backbone (Grounding DINO) than prior work and the increased computational latency. During the rebuttal period, the author addressed these concerns. They provided latency analyses, explaining the overhead as a common trade-off in TTA and demonstrating the method's strength over a baseline that uses the same detector. They also added new experiments comparing against other fine-tuning methods and analyzing hyperparameter sensitivity, which satisfied the reviewers. Therefore, AC recommends accept.